# Emotional Reaction to the First Dose of COVID-19 Vaccine: Postvaccination Decline in Anxiety and Stress among Anxious Individuals and Increase among Individuals with Normal Prevaccination Anxiety Levels

**DOI:** 10.3390/jpm12060912

**Published:** 2022-05-31

**Authors:** Rasmieh Al-Amer, Malakeh Z. Malak, Hala Mohammad Ramadan Burqan, Elena Stănculescu, Sylivia Nalubega, Abdulmajeed A. Alkhamees, Amin Omar Hendawy, Amira Mohammed Ali

**Affiliations:** 1Faculty of Nursing, Isra University, Amman 11953, Jordan; r.al-amer@outlook.com; 2School of Nursing and Midwifery, Western Sydney University, Penrith, NSW 2751, Australia; 3Community Health Nursing, Faculty of Nursing, Al-Zaytoonah University of Jordan, Amman 11733, Jordan; malakeh.m@zuj.edu.jo; 4School of Nursing, Alghad International Colleges for Applied Medical Sciences, Riyadh Branch, Riyadh 13315, Saudi Arabia; halaburqan991@gmail.com; 5Faculty of Psychology and Educational Sciences, University of Bucharest, 050663 Bucharest, Romania; elena.stanculescu@fpse.unibuc.ro; 6Department of Nursing, School of Health Sciences, Soroti University, Soroti City 211, Uganda; syliviaogwang@yahoo.com; 7Department of Medicine, Unayzah College of Medicine and Medical Sciences, Qassim University, Unayzah 52571, Saudi Arabia; 8Department of Animal and Poultry Production, Faculty of Agriculture, Damanhour University, Damanhour 22516, Egypt; amin.hendawy@gmail.com; 9Department of Psychiatric Nursing and Mental Health, Faculty of Nursing, Alexandria University, Smouha, Alexandria 21527, Egypt; mercy.ofheaven2000@gmail.com

**Keywords:** COVID-19, vaccine/vaccination, stress, anxiety, hesitancy, preparedness, Jordan/Arab

## Abstract

Although vaccination has been adopted by the WHO to limit worldwide transmission of COVID-19, people’s worries about COVID-19 vaccines may suppress their desire for vaccination despite vaccine availability. This study aimed to investigate anxiety and stress symptoms among 250 Jordanians (mean age = 43.18 ± 6.34 years, 72% females) who received their first vaccine dose. The respondents completed the anxiety and stress subscales of the Depression Anxiety and Stress scale 21 (DASS-21) before and after vaccination. The respondents expressed more moderate–severe levels of stress before than after vaccination (20.8% and 13.2%, respectively). Meanwhile, 37.2% and 45.2% of the respondents expressed moderate–severe anxiety before and after vaccination, respectively. Wilcoxon signed-rank test revealed that the drop in the level of stress from before- (median (IQR) = 5 (1–8)) to after vaccination (median (IQR) = 3 (1–7)) was statistically significant (z = −3.81, *p* = 0.001, r = 0.17) while the increase in anxiety was not. Anxiety significantly dropped postvaccination among individuals experiencing mild to severe anxiety before vaccination. Similarly, stress and anxiety significantly increased among individuals expressing normal anxiety before vaccination (z = −3.57 and −8.24, *p* values = 0.001, r = 0.16 and 0.37, respectively). Age positively correlated with postvaccination anxiety among respondents with mild prevaccination anxiety, and it negatively correlated with the prevaccination level of stress in the normal-anxiety group. Gender, marital status, respondents’ level of education, and history of COVID-19 infection had no significant correlation with anxiety or stress at either point of measurement. Overcoming their hesitancy to receive COVID-19 vaccines, individuals with normal levels of anxiety experienced a rise in their distress symptoms following immunization. On the contrary, vaccination seemed to desensitize anxious individuals. Policymakers need to formulate a population-specific plan to increase vaccine preparedness and promote psychological well-being over all during the pandemic.

## 1. Introduction

Coronavirus disease 2019 (COVID-19) is caused by the novel coronavirus strain “SARS-CoV-2”. Currently, as of May 2022, COVID-19 continues to have a trail of drastic negative effects on a large number of people around the world. This is because SARS-CoV-2 and its variants, Delta and Omicron, represent a highly contagious airborne infection, which spreads mainly through minute respiratory droplets or aerosols during close human contact, particularly while coughing and sneezing [1,2]. Since the commencement of the pandemic, the World Health Organization (WHO) has recommended a wide range of safety measures such as hand washing, wearing a face mask, maintaining social distancing, and banning large social gatherings [2,3]. All of these measures fall short in combating the spread of this serious and highly contagious viral infection [4]. Accordingly, the WHO approved COVID-19 vaccines on 31 December 2020. The strategy of vaccinating the world—inoculating 70% of all the world population by mid-2022—was adopted as a promising method to limit the pandemic [5].

Vaccination seems to be the best available option for fighting off the disease [6]. In this respect, unvaccinated individuals are reported to be 14 and 68 times more likely to die from COVID-19-related complications than those who are vaccinated and boosted, respectively [7,8]. Therefore, it is imperative to evoke a communal response and a sense of national purpose with the goal of combating the risk of human exposure to the virus. In other words, for the vaccine to be truly successful, the world population needs to accept and receive it [7].

The development of the vaccine was uncertain given multiple mutations in the viral genetic structure [2]. Moreover, the rushed development of the vaccine and its rapid availability to the global population had implications for the psychological state of some individuals. Lay people, even those who view vaccination as protective against COVID-19, feel anxious about the possibilities and risks of taking the vaccine [9,10,11]. Unfortunately, a significant number of people around the world still refuse to receive the vaccine [5,12]. Vaccination hesitancy is evoked by uncertainty and perceptual disintegration as a result of misinformation and conspiracy beliefs [13]. Misinformation on how COVID-19 vaccines are developed and tested, as well as on their safety and efficacy, is widely communicated through social media [9,14]. The influence of the globalized antivax movement on increased vaccine hesitancy is largely expressed through social media [5,9]. As a result of the widely communicated erroneous information about COVID-19 (e.g., it is a man-made disease and vaccines are intended to cause death in certain groups), vaccine hesitancy represents a growing obstacle, which may hinder the containment of the pandemic [3,5].

Anxiety and fear of COVID-19 are reported to increase intention to get vaccinated [13,15]. In fact, perceived vulnerability to COVID-19 is associated with more willingness to take the vaccine, while lower perceived vulnerability is associated with more vaccine hesitancy [14,15,16]. Nonetheless, the intention to get vaccinated decreases when anxiety and fear of COVID-19 are associated with high levels of existential anxiety, and this effect is mediated by conspiracy beliefs [13,15]. Indeed, those who market conspiracy beliefs tend to express greater fear and greater tendency toward psychopathology. They employ conspiracy beliefs as a method of coping with uncertainty surrounding the pandemic [3,17]. Therefore, vaccine hesitancy is largely attributed to negative emotions, which are accelerated by antivax rumors about COVID-19 vaccines [5,15].

Vaccine-related psychological concerns are probably underpinned by the unforeseen consequences and mistrust of those in charge, which in turn leads to further stress, anxiety, depression, and other psychological difficulties [13,14,18,19]. This notion is better understood within the frame of vulnerability postvaccination. Before the emergence of the Omicron variant, vaccines were stated to provide around 90% protection against COVID-19. However, it turned out that even with two doses of the vaccine, the efficiency dropped to approximately 50% protection within a few months of the second shot. In the wake of the Omicron variant, those who were fully vaccinated with two shots and an additional third booster shot, still found themselves subject to infection and unfortunately still managed to succumb to the Omicron variant [18,19]. Additionally, longitudinal investigations (six months after vaccination) show that compared with unvaccinated COVID-19 patients, patients receiving COVID-19 vaccination do not demonstrate improved post-acute sequelae of COVID-19 such as anosmia, respiratory symptoms (e.g., cough, dyspnea, phlegm, wheezing), depression, anxiety, post-traumatic stress disorder related to COVID-19 and other trauma, and quality-of-life [20].

Vaccine-related anxiety may also be triggered by a reported cluster of anxiety-related adverse events, which take place after the administration of COVID-19 vaccines [21,22]. Sudden rise in blood pressure following vaccination may raise uncertainties, since it is a typical symptom of both pseudo-allergy and anxiety [12]. In fact, high anxiety among adults attending COVID-19 vaccination centers is associated with fear that terrible consequences may happen [6,23,24]. An immunization-stress-related response may develop in people receiving the first as well as the second dose of the vaccine [25]. For some, vaccine fear and anxiety may be justifiable given that vaccine adverse effects can be severe in a trivial number of people (e.g., anaphylactic shock requiring resuscitation) [26,27]. Most events of allergic anaphylaxis occur in individuals with a prior history of allergy such as food allergy and allergy to wasp stings [26,27]. Animal studies report higher occurrence of such anaphylaxis and relate it to C-activation [12].

Negative emotional reactions (e.g., depression or anxiety) are recorded as adverse effects after COVID-19 vaccination [22,28,29]. In particular, anxiety-related events were reported among 8500 Janssen COVID-19 vaccine recipients, with up to 8.2 episodes per 100,000 doses as announced by the WHO. Among individuals expressing emotional/neurological symptoms after the COVID-19 vaccination, 18F-FDG PET/MRI scans revealed hypometabolism in the bilateral parietal lobes. These areas play a role in the fear network model that has been implicated in anxiety, which presents an empirical support of the immunization-stress-related response [28].

Aiming to vaccinate 80 percent of the population, the Jordanian Ministry of Health and the National Center for Crisis and Crisis Management launched the country’s largest-ever mass immunization campaign. At the beginning of the vaccination campaign, the country’s tally of COVID-19 cases reached 309,846, out of which 4076 resulted in death and 292,104 recovered [30]. Therefore, vulnerable individuals (healthcare workers and the elderly) were prioritized as the first to take the vaccine [31]. Generally, all Jordanian citizens are required to register online to be scheduled for COVID-19 vaccines. The government also gave the right for anyone living in Jordan to approach any of the 29 health centers in the kingdom to take their shots, even if they opted for a drive-through vaccination approach. Like other countries around the world, the local government in Jordan has pushed residents to receive COVID-19 vaccines through a wide range of actions. For example, vaccination records are authenticated through mobile and national identification numbers. Individuals have to show their vaccination records when they need access to public places (e.g., shopping malls and educational facilities). Those who totally refuse to receive the vaccine are asked to show a negative PCR test each week, which in itself is a financial, physical, and psychological burden. Since psychological reactions affect intentions to receive the vaccine, and they may stem as possible adverse effects of COVID-19 vaccines, this study opted to assess anxiety and stress symptoms before and 15 minutes after taking the first shot of the COVID-19 vaccine in a Jordanian community sample. According to the aforementioned literature, we hypothesized that the levels of stress and anxiety may increase from before to after vaccination.

## 2. Materials and Methods

### 2.1. Design, Setting, and Sampling

A longitudinal design was utilized to perform this study. A convenient sample was obtained through sequential recruitment of all individuals attending a vaccination center in the eastern part of the Jordanian capital (Amman) who agreed to take part in the study. A power analysis was performed in G* power version 3.1 [32] to calculate the needed sample size based on effect size = 0.2, power = 0.85, and alpha = 0.05. A sample size of 227 participants was required to successfully conduct the study; however, the questionnaire was administered to 260 respondents in order to make up for missing data. Participants were eligible to take part in this study if they met the following criteria: aged 18 years and over, able to read and write, and willing to participate in the study.

### 2.2. Measurements

The questionnaire used to conduct the current survey included three sections. Section one comprised questions on history of exposure to COVID-19 infection and the socio-demographic characteristics of the respondents: age, gender, marital status, and educational level.

Section two comprised the anxiety and stress subscales of the Depression Anxiety Stress Scale 21 (DASS-21). The DASS-21 was developed by Lovibond (1995). Compared with stand-alone measures of depression or anxiety, the DASS-21 is a single measure, which assesses three distinct mental symptoms: depression, anxiety, and stress [33]. The scale does not contain items pertaining to non-specific symptoms (e.g., loss of appetite), which exist in other measures such as the Beck Depression Inventory [34,35]. Additionally, the overall score of the DASS-21 can be used to reflect overall psychological distress [36]. Each subscale consists of seven items. Item responses are rated on a 4-point scale ranging from 0 (not applicable) to 3 (applies to me most of the time). The total scores of each subscale range between zero and 21. Individual scores can be classified to reflect normal, mild, moderate, severe, and extremely severe levels of the symptoms based on known cutoff points, which are reported below in Table 1 [33,34]. The Arabic version of the DASS-21 was adopted in this study [35]. This Arabic version has strong psychometric properties, with a Cronbach’s alpha of 0.95 [35,37]. The reliability of the stress and anxiety subscales in the current sample was excellent both before (Cronbach’s alpha = 0.91 and 0.89) and after vaccination (Cronbach’s alpha of both subscales = 0.90). Section three comprised a visual analog scale with scores from zero to ten to examine postvaccination pain at the site of injection.

### 2.3. Data Collection Procedure and Ethical Considerations 

The questionnaires were distributed and collected in the waiting room of a vaccination health center during the period between August 3 and August 10, 2021. Potential respondents were given a cover sheet introducing the purpose of the study and a consent form detailing participants rights (voluntary participation, data security, etc.), which they had to sign before completing the questionnaire. They were also informed that we would need them to fill in an identical questionnaire 15 minutes after receiving the shot. Ethical approval for this study was obtained from the Institutional Review Board (IRB) of Isra university (No. SREC/21/08/014). Data were handled according to the Jordanian standards of data protection.

### 2.4. Statistical Analysis

Ten respondents were excluded from the analysis because of incomplete responses—response rate = 96.1%. Based on normality tests, the anxiety and stress variables were presented using median and interquartile range (IQR), while the descriptive statistics of categorical variables were reported as frequency and percentage. Wilcoxon signed-rank test was used to examine the difference between the pre- and postvaccination levels of anxiety and stress in the whole sample. The effect size was estimated by dividing the z scores by the square root of the number of observations in pre- and postvaccination measurements. Because a considerable proportion of the respondents had normal or mild levels of anxiety before vaccination, we wanted to examine whether anxiety and stress scores change according to respondents’ prevaccination levels of anxiety. Therefore, the normal, mild, and moderate–severe anxiety subsamples were derived based on the cutoff scores of the anxiety subscale, which are shown in Table 1. Wilcoxon signed-rank test was used to examine the difference between the pre- and postvaccination levels of stress and anxiety in these subsamples. Spearman’s r correlation was used to examine the association of anxiety and stress at both points of measurement with history of COVID-19 infection and the sociodemographic characteristics of the respondents. Mann–Whitney U test and Kruskal–Wallis test were used to examine the differences in anxiety and stress scores before and after vaccination across groups of age (25 years and below, above 25 years), gender, education, and marital status (see Table 2 for categories). Kruskal–Wallis test was used to examine whether anxiety and stress scores before or after vaccination had an effect on postvaccination injection-site pain. This variable was developed by recoding pain scores on the pain visual analog scale into three levels: mild (1–3), moderate (4–6), and severe (7–10). The analysis was performed in SPSS (IBM Corp, Armonk, NY, USA) version 24, and the findings were considered to be significant at *p* < 0.05 in two-tailed tests.

## 3. Results

As shown in Table 2, the majority of the respondents were females, married, and had higher than secondary educational level. A minority had previously contracted COVID-19 infection. The respondents in the normal-anxiety subsample (*n* = 126) had a mean age of 43.1 ± 6.9 years, were largely females (75.4%), were mostly married (69.8%), had education higher than a secondary degree (72.2%), and 6.3% of them had previously contracted COVID-19 infection.

The respondents in the mild-anxiety subsample (*n* = 31) had a mean age of 43.1 ± 4.3 years, were largely females (64.5%), were mostly married (77.4%), had education higher than a secondary degree (74.2%), and 16.1% of them had previously contracted COVID-19 infection.

The respondents in the moderate–severe anxiety subsample (*n* = 93) had a mean age of 43.3 ± 6.2 years, were largely females (69.9%), were mostly married (58.1%) or single (19.4%), had education higher than a secondary degree (68.8%), and 6.5% of them had previously contracted COVID-19 infection.

Mild pain severity after vaccination was expressed by 86.5%, 83.9%, and 74.2% of the respondents in the normal, mild, and moderate–severe anxiety groups, respectively.

Table 1 shows that 20.8% of the study participants had moderate to severe levels of stress before they were vaccinated. Severe and extremely severe forms of stress were reported by only 8.0% and 4.8% of the respondents, respectively. Remarkably, stress levels decreased after receiving a COVID-19 vaccine, with around 87% of the respondents reporting no stress and only 3.2% having a severe form of stress. Around 50% of the respondents experienced anxiety before vaccination, and approximately 18.8% experienced severe to extremely severe levels of anxiety. After vaccination, 61.2% of the respondents experienced anxiety, with 13.4% of them reporting a severe to extremely severe form of anxiety.

Table 3 shows a significant decrease in stress scores in the whole sample from before to after vaccination, with a small effect size (r = 0.17). On the contrary, the anxiety level increased from before to after vaccination, albeit that change was non-significant. Consistent with our hypothesis, the levels of stress and anxiety significantly increased in the normal-anxiety group after vaccination compared to before they received the vaccine. The effect size for the change in stress was small, but it was moderate for anxiety (Table 3). Unexpectedly, the levels of stress and anxiety dropped in the mild-anxiety group, albeit the change was significant only in the anxiety median. In the moderate–severe anxiety subsample, the Wilcoxon signed-rank test revealed a significant reduction in the levels of stress and anxiety from before to after vaccination, and the effect size was moderately strong (r = 0.32 and 0.33).

Among all sociodemographic characteristics, the prevaccination level of stress was significantly associated only with age in the whole sample (r = −0.154, *p =* 0.015) and the normal-anxiety subsample (r = −0.250, *p =* 0.005). Age was positively associated with anxiety after vaccination in the mild-anxiety subsample (r = 0.383, *p =* 0.033). However, Mann–Whitney U test and Kruskal–Wallis test revealed no significant contribution of any of the sociodemographic characteristics or previous history of COVID-19 infection to the scores of stress or anxiety at either point of measurement in the whole sample and the normal-anxiety subsample (all *p* values > 0.05, see Appendix A for details of test results). Prevaccination anxiety scores were significantly higher among respondents aged 25 years or below in the moderate–severe anxiety subsample (U = 630.0, z = −2.12, *p =* 0.034). Prevaccination stress scores contributed to postvaccination pain severity at the injection site in the entire sample (H (2) = 6.98, *p =* 0.031) and in the moderate–severe anxiety subsample (H (2) = 9.88, *p =* 0.007).

## 4. Discussion

The success of the wide-scale vaccine campaigns in all countries of the world is highly dependent on vaccine acceptance by the majority of the population [7]. The current study examined pre- to postvaccination change in the levels of anxiety and stress symptoms in a sample of Jordanians who received their first dose of COVID-19 vaccine. Consistent with our hypothesis, both anxiety and stress significantly increased after vaccination among individuals experiencing a normal prevaccination level of anxiety. On the contrary, the levels of stress significantly decreased in the entire sample after vaccination. In the meantime, the levels of both stress and anxiety dropped after vaccination among those with prevaccination anxiety, and that drop was more pronounced among individuals with moderate–severe anxiety than in those with mild anxiety.

Our results are consistent with those of some previous studies. Mild anxiety was reported among 76.6% of Indonesian adults receiving their first shot [23]. Vaccine-related distress symptoms were reported to be high among Chinese adults, but their levels significantly dropped following vaccination [24]. Anxiety is reported to contribute to the acceptance and willingness to receive COVID-19 vaccine. In this context, Chinese patients with a formal psychiatric diagnosis (major depression or anxiety disorder) were more willing to receive and pay for the COVID-19 vaccine than were healthy individuals [38]. In line with a large-scale study comprising an analysis of health records and a survey of psychiatric patients, patients with substance and tobacco abuse disorders significantly expressed higher vaccine hesitancy than patients with all anxiety disorders (e.g., generalized anxiety and post-traumatic stress disorder) and major depression [39]. In an Italian sample surveyed during the lockdown period in 2020, anxiety and death anxiety had a direct positive effect on the propensity to receive the vaccine [13].

Anxiety may result from high levels of perceived susceptibility to COVID-19. Perceived vulnerability and use of the vaccine for self-protection are associated with increased vaccine acceptance in China and the United Kingdom [10,14,16]. The effect of anxiety on COVID-19 vaccine acceptance may vary as a function of concerns about vaccine safety and effectiveness, which are evidently prompted by the novelty and rapid development of the vaccine [6,10,13,14]. These concerns may be largely shaped by misinformation circulated through social media. In fact, research relates social media, as the most trusted information source, to increased parental vaccine hesitancy compared with trusted official information sources [24,40]. Verbal reports of vaccine recipients indicate that anxiety is associated with fear of the development of serious adverse effects [6,23]. Accordingly, the noticed reduction in stress and anxiety symptoms among the anxious subset of our sample may be the result of failure of participants’ priori expectations of developing serious adverse effects after receiving the vaccine—a reassuring effect of their positive experience with the vaccine.

Contrary to our results, high anxiety is associated with decreased intention to receive the vaccine in some groups. Among healthcare workers, the occurrence of an immunization-stress-related response was significantly higher among those with strong prevaccination anxiety and history of allergy [25]. Anxiety symptoms among Chinese psoriatic patients were associated with high vaccine hesitancy. A slight to significant deterioration in psoriasis is reported in 20% of patients who experienced stress, anxiety, and depression after vaccination [41]. These reports may be justified by interactions taking place between anxiety and other factors. For example, death anxiety increased the tendency to take the vaccine in an Italian survey. However, it reduced the propensity to get vaccinated through a mediated path in believing in conspiracy theories, whereas paranoia was linked to a reduction in vaccination adherence with the mediation effect of mistrust in medical science [13]. Likewise, a longitudinal study involving a large representative sample from the U.S. documents reduction in mental distress symptoms in bi-weekly and four-week survey cycles. This study indicates a persistent decline in mental distress following vaccination [42]. Reports involving adults who received booster doses of the COVID-19 vaccine associate the greatest levels of anxiety and mistrust with the first dose compared with later doses [43]. Distress symptoms also decreased after receiving the vaccine in China. However, those symptoms remained persistently elevated after receiving the vaccine among those with mistrust in vaccine efficacy and history of vaccine-related allergic reactions [24]. Experimental evidence indicates that a long-lasting increase in anxiety-like behaviors following stressful events is a function of pre-existing alterations in corticosteroid signaling [44]. Given a lack of assessment before vaccination, structural brain alterations detected in individuals with anxiety after the COVID-19 vaccine may be associated with the previous accumulation of physiological and structural adversities, which are further enhanced by vaccine uncertainty in certain groups [28]. Given the role of psychological and physical vulnerability (e.g., mistrust, allergy, chronic diseases) to distress following COVID-19 vaccine [24,41], it is necessary to identify and properly manage groups liable to high levels of psychological distress following vaccination to alleviate psychological and physical morbidity, which may ensue as a result.

In our analysis, the prevaccination level of stress was associated with more pain at the injection site 15 minutes after vaccination in the whole sample, and more evidently in the moderetae-severe anxiety subsample. Local side effects after COVID-19 vaccines are common, and they have been reported in 79% of Jordanian healthcare workers who received the vaccine [45]. Activation of the hypothalamic–adrenal axis in stressful conditions is associated with modulation of the immune response [46,47,48]. Modulation of stress signaling by drugs that regulate the activity of the hypothalamic–adrenal axis (e.g., corticosteroid) is involved in the mitigation of chronic pain (e.g., musculoskeletal and spinal pain). These drugs are reported to interfere with the activity of COVID-19 vaccines [49,50]. Additionally, alterations in humoral immunity are reported six months following COVID-19 vaccination among patients with inflammatory bowel disease who receive anti-TNF (infliximab, adalimumab) therapy [51]. Therefore, we might expect that high prevaccination level of stress might disrupt immune regulation, resulting in increased flux of immune cells and accelerated release of cytokines, chemokines, and reactive oxygen species at injection site, eventually leading to more local pain, redness, and swelling and probably more other adverse effects (e.g., generalized muscle ache) [52,53].

In our study, young age was associated with high prevaccination stress among individuals with normal anxiety, while older age was associated with high postvaccination stress in the mild-anxiety subsample. This result is in accordance with reports associating age with greater vaccine hesitancy and anxiety among Chinese population, psoriatic patients, and Indonesian adults [23,24,41]. Gender is known to exert significant effects on emotional processing and related emotional and behavioral consequences [54,55]. Gender and education are also reported to be associated with vaccine hesitancy/anxiety [6,23,24,40,56]. Being previously infected with COVID-19 is a reported predictor of vaccine hesitancy [24,39]. However, none of the respondent’s demographic characteristics or history of being infected with COVID-19 was associated with vaccine-related anxiety and stress in this study. This difference may be attributed to the high presentation of females and those with above high school level of education, as well as the small number of previous COVID-19 cases in our sample (Table 2).

### Strength, Implications, and Limitations

This study provides new information, denoting a desensitization effect of COVID-19 vaccine among anxious adults. The findings also show that adults with normal anxiety levels are more prone to develop distress symptoms following vaccination than their anxious counterparts. Moderate–severe anxiety before vaccination was higher among younger groups. Higher levels of prevaccination stress in individuals with moderate–severe anxiety before vaccination are associated with increased pain severity at the injection site. Because the study did not follow up the respondents for a long time after receiving the vaccine, the nature of subsequent changes in the symptoms of distress and local pain in different groups is unclear. This issue needs to be handled in future studies, given that elevated mental symptoms after COVID-19 vaccine may deteriorate pre-existing pathologies if these symptoms are not properly managed [41]. Policymakers should adopt strategies that may combat conspiracy theories and increase participants’ vaccine preparedness [5]. Access to trusted vaccine-related information is associated with reduced vaccine hesitancy and postvaccination distress [24]. Therefore, key strategies to promote vaccine acceptance should focus on the use of social media to disseminate correct information about the vaccine as well as hotline services to provide postvaccination follow up and support for vaccine recipients [57,58]. Youth represent an important target group for infodemic campaigns, given their high use of social media as a source of information and method of socialization during the pandemic [24,54,59]. Gaming has been recommended by the WHO as a possible method for maintaining social interactions during COVID-19 lockdown. Moreover, some games (e.g., Go Viral!) have been introduced and are reported to have reduced the perceived reliability of fake vaccine news by an average of 21% [57,60].

The use of a convenient sample from a single healthcare center is another limitation, which may limit the generalizability of the results. In addition, we were not able to identify factors associated with the change in mental symptoms because of lack of assessment of other variables, which may explain the noticed changes such as COVID-19 fear/anxiety/trauma, perceived susceptibility, conspiracy beliefs, and sources of information on COVID-19 vaccine. The results of a nationally representative survey in the United Kingdom show that individuals are more willing to receive fictitious vaccines, which are putatively produced by the United States government Medicare program, than to take the Sputnik and Sinovac vaccines, which are produced by the Russian and Chinese governments [61]. Likewise, Italian adults expressed more trust in and less fear of Pfizer-BioNTech compared with Vaxzevria (AstraZeneca) [43]. Regrettably, data on the types of COVID-19 vaccines received by our respondents were not assessed. Accordingly, this study fails to detect differences in the emotional reaction to vaccination among the recipients of different COVID-19 vaccines.

It is well known now that obesity and chronic physical disorders increase the risk for COVID-19 and its complications [2]. Patients with these characteristics may express higher anxiety and stress than other groups, irrespective of the vaccine [3]. These characteristics may also be related to the negative emotional reactions associated with COVID-19 vaccine, though we could not locate any study investigating such relationship. Unfortunately, we did not collect data on the anthropometric and clinical characteristics of the respondents, which may indicate a possibility of bias of the reported results by such characteristics. The number of respondents in some of the subsamples was small (e.g., the mild-anxiety subsample), indicating insufficient power for analyses performed in these subsamples. The credibility of the results would be better if the levels of vaccine distress symptoms were also examined in a comparison group of unvaccinated individuals, which was not applicable in the current study. Moreover, the data were collected in 2020, and the pandemic is ongoing. Therefore, more studies are needed to examine the stability of the findings over time.

## 5. Conclusions

The COVID-19 vaccine represents a source of distress for first-dose recipients, with increased levels of stress and anxiety after vaccination among those with normal levels of anxiety before vaccination. The COVID-19 vaccine seems to exert a desensitization effect among anxious individuals. Further investigations of the dynamics of the change in distress symptoms in different groups are needed.

## Figures and Tables

**Table 1 jpm-12-00912-t001:** Levels of stress and anxiety before and after administering the COVID-19 vaccine (*n* = 250).

Variable	Before Vaccination *n* (%)	After Vaccination *n* (%)
**Stress**		
No stress = 0–7	185 (74.0)	217 (86.8)
Mild = 8–9	13 (5.2)	12 (4.8)
Moderate = 10–12	20 (8.0)	13 (5.2)
Severe = 13–16	20 (8.0)	8 (3.2)
Extremely severe ≥ 17	12 (4.8)	0 (0.0)
**Anxiety**		
Normal = 0–3	126 (50.4)	97 (38.8)
Mild = 4–5	31 (12.4)	40 (16.0)
Moderate = 6–7	46 (18.4)	82 (32.8)
Severe = 8–9	10 (4.0)	17 (6.8)
Extremely severe ≥ 10	37 (14.8)	14 (5.6)

*n*: number, %: percentage.

**Table 2 jpm-12-00912-t002:** Demographic characteristics of the participants (*n* = 250).

Characteristics of the Respondents	*n* (%)
**Gender**	
Male	70 (28)
Female	180 (72)
**Marital Status**	
Single	46 (18.4)
Married	166 (66.4)
Divorce	15 (6.0)
Widow	23 (9.2)
**Educational Level**	
Secondary and less	72 (28.8)
Higher than secondary	178 (71.2)
**Age in Years**	
Mean (SD)	43.18 (6.34)
Range	18–63
**COVID-19 Infection History**	
Yes	19 (7.6)
No	231 (92.4)
**Pain at Injection Site**	
Mild	204 (81.6)
Moderate	40 (16.0)
Severe	6 (2.4)

*n*: number, %: percentage, SD: standard deviation.

**Table 3 jpm-12-00912-t003:** Descriptive statistics of stress and anxiety symptoms among the respondents and differences in the levels of stress and anxiety before and after the administration of the first dose COVID-19 vaccine.

Variables	Samples	MD (IQR) before Vaccination	MD (IQR) after Vaccination	z of Wilcoxon Signed-Rank Test	*p*	r
Stress	Whole sample (*n* = 250)	5.0 (1.0–8.0)	4.0 (1.0–7.0)	−3.81	0.001	0.17
Anxiety		3.0 (1.0–7.0)	5.0 (1.0–7.0)	−0.53	0.597	0.02
Stress	Normal anxiety (*n* = 126)	2.0 (0.0–4.0)	4.5 (0.0–6.3)	−3.57	0.001	0.16
Anxiety		1.0 (0.0–2.0)	5.0 (0.0–7.0)	−8.24	0.001	0.37
Stress	Mild anxiety (*n* = 31)	4.0 (3.0–7.0)	3.0 (3.0–5.0)	−1.28	0.201	0.06
Anxiety		4.0 (4.0–5.0)	3.0 (3.0–5.0)	−2.70	0.007	0.12
Stress	Moderate–severe anxiety (*n* = 93)	8.0 (7.0–14.0)	5.0 (2.0–8.0)	−7.13	0.001	0.32
Anxiety		8.0 (7.0–13.0)	6.0 (1.0–8.0)	−7.27	0.001	0.33

MD: median; IQR: interquartile range; r: reflects effect size. N.B. Groups are described based on prevaccine levels of anxiety on the anxiety subscale of the Depression Anxiety Stress Scale 21.

## Data Availability

The dataset used to produce the current study [62] is available in Mendeley at https://data.mendeley.com/datasets/b4pdcc4mh4/1 (accessed on 24 May 2022).

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
