# Peer review of "Emotional Reaction to the First Dose of COVID-19 Vaccine: Postvaccination Decline in Anxiety and Stress among Anxious Individuals and Increase among Individuals with Normal Prevaccination Anxiety Levels"

_jpm, 2022, doi:10.3390/jpm12060912_

Round 1

Reviewer 1 Report

Journal: Journal of Personalized Medicine; Manuscript Number: jpm-1735618

Authors: Rasmieh Al-Amer et al.

Title: "Emotional reaction to the first dose of COVID-19 vaccine: Post vaccination decline in anxiety and stress among anxious individuals and increase among individuals with normal anxiety levels pre vaccination."

The authors of the manuscript tried to evaluate anxiety and stress levels among individuals after their first dose of the COVID-19 vaccine. According to the authors, the levels of stress significantly reduced after vaccination. Moreover, anxiety levels decreased among anxious participants (i.e., mild to severe anxiety levels before vaccination). On the other hand, stress and anxiety levels increased among participants with normal pre-vaccination anxiety levels. The following points should be considered.

Major Comments

  1. According to the authors, the needed sample size was 227 participants. However, the study population was divided into three main subgroups. Was the sample size in these groups appropriate to have sufficient power?
  2. Please define if written consent was obtained from the participants.
  3. It would be helpful if the authors could include further information regarding the anthropometric and clinical characteristics of the participants and the presence of risk factors associated with COVID-19. Furthermore, it would be important if the investigators could consider the effect of these variables on anxiety and stress levels and the related outcomes.
  4. The majority of the participants is females. Did the study findings differ among females and males?
  5. Did the authors consider whether their findings differ among the different COVID-19 vaccines?

Minor Comments

  1. It should be helpful if the authors could add further descriptions or legends for each table included in the supplementary material, organize them in a more comprehensive way, and keep only the necessary information.

Author Response

Journal: Journal of Personalized Medicine; Manuscript Number: jpm-1735618

Response to the comments of Reviewer 1

Dear Reviewer 1,

Greetings!!

We are grateful to the reviewer for allowing us a chance to improve the manuscript. We have modified the manuscript taking all these valuable and directive comments into consideration. Authors’ responses come underneath in red.

Major Comments

  1. According to the authors, the needed sample size was 227 participants. However, the study population was divided into three main subgroups. Was the sample size in these groups appropriate to have sufficient power?

Thank you for raising this important issue. We have not conducted power analysis for subsamples because they were created post hoc (after data collection) based on symptoms cut-off scores expressed by the entire sample. However, we agree with the reviewer that the number of respondents in some subsamples (e.g., mild anxiety) was small, which may affect the credibility of the results. We have noted that in the limitation section in this version (line 402-404).

  1. Please define if written consent was obtained from the participants.

Yes, a written consent was obtained from the participants. This has been clarified in the current version (line 187-189).

  1. It would be helpful if the authors could include further information regarding the anthropometric and clinical characteristics of the participants and the presence of risk factors associated with COVID-19. Furthermore, it would be important if the investigators could consider the effect of these variables on anxiety and stress levels and the related outcomes.

It is well-known now that obesity and chronic physical disorders are at greater risk for covid-19 and its complications. Patients with these characteristics may express higher anxiety and stress than other groups, irrespective of the vaccine. These characteristics may also be related to the emotional reaction to the vaccine. Unfortunately, we have not collected data on anthropometric and clinical characteristics of the respondents, which may bias the results. We have noted this as a limitation (line 396-402).

  1. The majority of the participants is females. Did the study findings differ among females and males?

Yes, we agree with the reviewer, results may be affected by gender heterogeneity. Nonetheless, our results did not differ across men and women. To avoid inflating the text with reports on non-significant tests, we have reported p values (> 0.05) in the current version and sufficiently clarified that the detail of this result is shown in the supplementary materials (line 262-267).

  1. Did the authors consider whether their findings differ among the different COVID-19 vaccines?

 Regrettably, data on the types of COVID-19 vaccines were not assessed. Accordingly, it has been included as another reported limitation in this revised version (line 389-395).

Minor Comments

  1. It should be helpful if the authors could add further descriptions or legends for each table included in the supplementary material, organize them in a more comprehensive way, and keep only the necessary information.

Yes, we have removed the results, which are reported in the manuscript. Only results that are not reported but referred to in the text are retained in the supplementary file. Titles have been included on top of the supplementary tables, and key variables were highlighted in yellow for better readability. Thank you so much.

We hope that the manuscript has been satisfactorily modified and that the current version will be suitable for publication.

Best regards,

Reviewer 2 Report

In the manuscript authors presented findings on the anxiety level associated with receiving the first dose of COVID-19 vaccine. The topic is interesting, but the major concern is measuring of "basal" or "prevaccination" level of anxiety in the waiting room while subjects were waiting for vaccine shot. As authors commented in the Discussion section, anxiety has been found to be associated even with intention to receive the vaccine. On the other side, 15 min after vaccination is a brief period for full anxiety spectrum symptomatology development, and many serious and important symptoms arising from delayed wariness, expectations and beliefs could have escaped from this time window.

Line 153: Authors stated: " A convenience sampling technique was used to recruit individuals....". Please clarify "a convenience sampling technique".

Why did authors decide to use DASS-21 for anxiety measurement? What are the advantages of that scale over e.g., Hamilton Anxiety Rating Scale?

Even though anxiety level was found to be not associated with educational level, age, and gender in the study (as can be seen from supplementary data provided), it is worthy to mention that in the text.

Author Response

Journal: Journal of Personalized Medicine; Manuscript Number: jpm-1735618

Response to the comments of Reviewer 2

Dear Reviewer 2,

Greetings!!

Yes, thank you so much for the reviewer’s sincere efforts, time, and patience as well as the insightful comments. We admit that the past version had many flaws, and we are keen to modify the manuscript properly. So, we have addressed all the comments meticulously. Authors’ responses to the comments come underneath in red.

In the manuscript authors presented findings on the anxiety level associated with receiving the first dose of COVID-19 vaccine. The topic is interesting, but the major concern is measuring of "basal" or "prevaccination" level of anxiety in the waiting room while subjects were waiting for vaccine shot. As authors commented in the Discussion section, anxiety has been found to be associated even with intention to receive the vaccine. On the other side, 15 min after vaccination is a brief period for full anxiety spectrum symptomatology development, and many serious and important symptoms arising from delayed wariness, expectations and beliefs could have escaped from this time window.

Yes, the reviewer is absolutely correct. This point represents a major limitation, which we have explicitly indicated so that the readers would be cautious about the results.

Line 153: Authors stated: " A convenience sampling technique was used to recruit individuals....". Please clarify "a convenience sampling technique".

Yes, this section has been further clarified, indicating sequential involvement of respondents who were willing to participate in the study (line 153-155).

Why did authors decide to use DASS-21 for anxiety measurement? What are the advantages of that scale over e.g., Hamilton Anxiety Rating Scale?

Yes, there are many scales for measuring anxiety. Because the readers may have the same question, we have noted in the text that the DASS-21 is a brief measure, which is used to evaluate three specific mental symptoms: depression, anxiety, and stress. Moreover, the overall score of the DASS-21 is used as an indicator of psychological distress (line 167-172).

Even though anxiety level was found to be not associated with educational level, age, and gender in the study (as can be seen from supplementary data provided), it is worthy to mention that in the text.

Yes, as the reviewer indicated in this comment, most of the sociodemographic characteristics had no contribution to anxiety and stress pre and post vaccination. We have briefly indicated this in the Results (line 252-257) and spotted it once again in the Discussion (line 338-350). Because the results of the relevant tests were non-significant, we included the related results as a supplementary file, with adequate reference given in the manuscript (line 256-257). This is primarily to prevent text inflation with unnecessary details. Because some readers may be interested in referring to test results, the supplementary file has been further refined in this revision to allow better access to and readability of the results: 1) deleting unnecessary results, which are reported in the manuscript such as Wilcoxon test, 2) ordering all results in the whole sample and three subsamples in the same sequence, and 3) including titles of supplementary tables and highlighting key variables in those tables.

In this revision, we have checked the manuscript for typos and other linguistic issues. Thank you once again, and we hope that the current version is suitable for publication.

Best regards,

Round 2

Reviewer 2 Report

Authors revised manuscript according to the reviewer's comments, but it is needed to include more detailed discussion regrding the anxiety measurement only at the brief period after vaccination (which is the major limitation of the present study).

Author Response

Journal: Journal of Personalized Medicine; Manuscript Number: jpm-1735618

Response to the comments of Reviewer 2

Dear Reviewer 2,

Greetings!!

Yes, thank you so much for the reviewer for allowing us a second chance to improve the manuscript. The comment received is of a pivotal importance, and proper discussion of the topic is a must. Authors’ responses to the comments come underneath in red.

Authors revised manuscript according to the reviewer's comments, but it is needed to include more detailed discussion regrding the anxiety measurement only at the brief period after vaccination (which is the major limitation of the present study).

Yes, we agree with the reviewer that very short follow up may provide misleading results, and this is a key limitation in the current study. We have refined the keyword search and obtained two large-scale longitudinal studies, which report similar results both in the US and Italy. Thank you so much.

In this revision, we have checked the manuscript once more for typos and other linguistic issues. Thank you once again, and we hope that the current version is suitable for publication.

Best regards,
